# Anti-HIV-1 Effect of the Fluoroquinolone Enoxacin and Modulation of Pro-Viral hsa-miR-132 Processing in CEM-SS Cells

**DOI:** 10.3390/ncrna11010008

**Published:** 2025-01-20

**Authors:** Verena Schlösser, Helen Louise Lightfoot, Christine Leemann, Seyedeh Elnaz Banijamali, Aathma Merin Bejoy, Shashank Tiwari, Jeffrey L. Schloßhauer, Valentina Vongrad, Andreas Brunschweiger, Jonathan Hall, Karin J. Metzner, Jochen Imig

**Affiliations:** 1Institute of Pharmaceutical Sciences, ETH Zurich, 8093 Zurich, Switzerland; 2Division of Infectious Diseases and Hospital Epidemiology, University Hospital Zurich, University of Zurich, 8091 Zurich, Switzerland; 3Institute of Medical Virology, University of Zurich, 8057 Zurich, Switzerland; 4Max Planck Institute of Molecular Physiology, Chemical Genomics Centre, 44227 Dortmund, Germany; 5Department of Pharmaceutical and Medicinal Chemistry, University Würzburg, 97074 Würzburg, Germany

**Keywords:** HIV-1, miRNA, fluoroquinolones, Enoxacin, hsa-miR-132, RNA interference, DICER1 processing

## Abstract

Background: Despite tremendous advances in antiretroviral therapy (ART) against HIV-1 infections, no cure or vaccination is available. Therefore, discovering novel therapeutic strategies remains an urgent need. In that sense, miRNAs and miRNA therapeutics have moved intensively into the focus of recent HIV-1-related investigations. A strong reciprocal interdependence has been demonstrated between HIV-1 infection and changes of the intrinsic cellular miRNA milieu. This interrelationship may direct potential alterations of the host cells’ environment beneficial for the virus or its suppression of replication. Whether this tightly balanced and controlled battle can be exploited therapeutically remains to be further addressed. In this context, the fluoroquinolone antibiotic Enoxacin has been demonstrated as a potent modulator of miRNA processing. Here, we test the hypothesis that this applies also to selected HIV-1-related miRNAs. Methods: We studied the effect of Enoxacin on HIV-1 replication coupled with miRNA qRT-PCR analysis of HIV-1-related miRNAs in CEM-SS and MT-4 T-cells. The effects of miRNA mimic transfections combined with Enoxacin treatment on HIV-1 replication were assessed. Finally, we employed an in vitro DICER1 cleavage assay to study the effects of Enoxacin on a pro-HIV-1 miRNA hsa-miR-132 processing. Results: We established that Enoxacin, but not the structurally similar compound nalidixic acid, exhibits strong anti-HIV-1 effects in the T-cell line CEM-SS, but not MT-4. We provide experimental data that this effect of Enoxacin is partly attributed to the specific downregulation of mature hsa-miR-132-3p, but not other tested pro- or anti-HIV-1 miRNAs, which is likely due to affecting DICER1 processing. Conclusions: Our findings show an anti-retroviral activity of Enoxacin at least in part by downregulation of hsa-miR-132-3p, which may be relevant for future antiviral therapeutic applications by modulation of the RNA interference pathway.

## 1. Introduction

The human immunodeficiency virus type 1 (HIV-1), the causative agent of AIDS, exhibits a leading account of infectious disease death especially in southern African countries. It affects more than 38 million people worldwide, with about 1.7 million new infections in adults and more than 500.000 deaths per annum (status 2019) [1]. Despite tremendous success with HIV treatments using antiretroviral therapy (ART), affording a relatively normal life span expectation for HIV-1 patients, no broadly accessible cure or anti-viral vaccination/elimination strategy has been developed. Consequently, improvements in current treatments or developing novel therapeutic approaches remain an important aim. Nonetheless, the nature of the virus to develop a plethora of escape mechanisms against host immune responses or therapeutic interventions makes it a demanding task, particularly in the context of life-long treatment. One attractive strategy appears to be the therapeutic targeting of endogenous host factors essential for virus replication.

MicroRNAs (miRNAs) are negative regulators of post-transcriptional gene expression; hence, they fulfill pivotal roles in almost all known physiological or pathophysiological conditions, for example, in growth regulation, development, cancer and virus replication (reviewed in [2,3,4,5]). Single miRNAs act on many different messenger RNAs (mRNAs), which in turn can be regulated by different miRNAs in concert through a highly regulated and fine-tuned network to sustain robust gene expression profiles [6]. In particular, it has been shown that certain host miRNAs impact viral gene expression and consecutive replication [7,8,9,10]. Thereby, miRNAs may affect the virus directly, by targeting viral transcripts, or indirectly by targeting cellular factors required for the virus replication (reviewed in [11]). For example, the expression of the hepatocyte-specific host miRNA hsa-miR-122 is required for the efficient RNA genome replication of the human hepatitis C virus (HCV) [12]. Inhibition of hsa-miR-122 using the anti-miRNA oligonucleotide inhibitor miravirsen leads to repression of HCV virus load in HCV patients [13].

The identification of host miRNAs necessary for the HIV-1 life cycle [14] would be highly desirable for the development of novel, but analogous therapeutic agents. On this note, several groups demonstrated that hsa-miR-29a inhibits HIV-1 replication by directly binding and repressing the nef transcript in HEK293T and Jurkat T cells [11,15,16,17,18,19]. However, it was also reported that binding of hsa-miR-29a to HIV-1 transcripts may be impaired by its secondary structure [16,20]. Moreover, an increasing number of other host miRNAs play similar roles, targeting HIV-1 directly or indirectly to hamper with its viral life cycle (e.g., hsa-miR-150, hsa-miR-223, hsa-miR-28 or hsa-miR-125b and others [11,21]).

Several groups reported a selective modulation of the host miRNAs upon HIV-1 infection. However, this data is contentious partly because different experimental setups were employed such as timing, cell types, and virus strains. For instance, Huang et al. compared activated vs. resting CD4+ T-cells and noted an enrichment of some miRNAs that potentially target the 3‘ untranslated region (UTR) of the HIV-1 mRNA. Strikingly, inhibition of the very same miRNAs in ART-treated patient-derived resting CD4+ T-cells led to an increase in virus replication, thus suggesting their contribution to the maintenance of HIV-1 latency [21]. Using alternative methods and cellular systems, a number of studies have described additional miRNAs with the potential to target HIV-1 RNA, thereby impacting HIV-1 replication [16,22,23]. However, data from our laboratory and others using Ago-2 Photoactivatable Ribonucleoside Enhanced Crosslinking and Immunoprecipitation (PAR-CLIP) methods in HIV-1 infected T-cell lines or primary infected macrophages suggest that HIV-1 is probably not directly regulated by host miRNAs, nor does the virus express its own small RNAs for auto- or host cell regulation [20,24].

There are a few reported examples of host miRNAs that enhance replication of HIV-1; for instance, on the one hand hsa-miR-132-3p was found to reactivate the latent pro-virus and, on the other hand to enhance HIV-1 replication in activated CD4+ T-cells. This phenotype was potentially driven in part by targeting of MeCP2 (methyl CpG-binding protein 2) a context-dependent transcriptional repressor or activator [25]. Similarly, a few other miRNAs relevant for targeting host genes were reported to restrict the HIV-1 life cycle [26]. For instance, hsa-miRs-34a-5p, -34c-5p, -217 and let-7 have been identified as additional pro-HIV-1 miRNAs, which enhance virus replication by targeting transcripts coding for the HIV-1 protein interactome [27,28,29,30,31]. These studies underline a tight balance and importance of host cell miRNAs for HIV-1 replication. It follows that, in analogy to hsa-miR-122, these miRNAs may serve as new therapeutic host-cell targets aggravating viral escape mechanisms. Therefore, a full understanding of the functions and the targetomes of these miRNAs are required in order to evaluate their potential as drug targets.

We searched for known examples of drug-like small molecules that might couple modulation of miRNA activity with an anti-HIV-1 capacity. Using this approach, some structures of class of fluoroquinolone antibiotics emerged due to their reported protection of HIV-1-induced cytopathic effects and inhibition of virus replication in MT-4 T-cells [32]. Enoxacin, a member of this compound class, did not show any protective effect in MT-4 cells, but showed a cytoprotective effect in the T-cell line CEM revealing that this effect of Enoxacin is cell-type dependent [33]. The recent intriguing discovery that Enoxacin modulated miRNA processing prompted us to investigate a potential anti-HIV-1 replicative effect in vitro in the context of miRNAs playing substantial roles in the HIV-1 life cycle [34,35,36,37,38,39,40,41,42]. Current reports documented that Enoxacin operates as an RNAi-enhancing agent by directly binding and activating TAR-binding protein (TRBP), an important co-factor required for correct DICER1 processing during miRNA biogenesis [34,43,44]. This trait of Enoxacin was observed to be not generic for all miRNAs, as miRNA maturation was also reduced in some cases, while a majority remained unchanged [34,42,45]. Additionally, Enoxacin is considered as a potential anti-cancer drug since it restored the expression of a subset of repressed miRNAs involved in tumorigenesis [45]. We hypothesized that the reported protective properties of some fluoroquinolones towards HIV-1 mediated cytopathy may be at least partially conferred by their RNAi enhancing potential; furthermore, they might affect the processing of the aforementioned pro- or anti-HIV-1-related miRNAs. In this work, we demonstrate that Enoxacin, but not the related fluoroquinolone nalidixic acid, strongly inhibits HIV-1 replication and that some HIV-1-related miRNAs were modulated by this compound in a cell type specific manner. More specifically, we show that Enoxacin negatively affects the processing of the pro-HIV-1 miRNA hsa-miR-132-3p contributing—at least partially—to its repression of HIV-1. Finally, we validate that pre-miR-132 maturation into both 3p and 5p strands by DICER1 is specifically decreased by Enoxacin due to induced structural re-arrangements in vitro.

## 2. Results

### 2.1. Expression Profiling of Pro- and Anti-HIV-1 miRNAs in Response to Enoxacin and Nalidixic Acid in Cellulo

To test our initial hypothesis, at first, we profiled the expression of selected HIV-1-related miRNAs and their response to the fluoroquinolones Enoxacin [34] and the structurally related nalidixic acid in HIV-1 infected leukemic cell lines CEM-SS and MT-4. We selected cell lines CEM-SS and MT-4 both derived from acute childhood or adult T lymphoblastic leukemia. Both CEM-SS and MT-4 cell lines have been described as suitable model systems for monitoring HIV-1 infection in vitro [46,47] as well as being already tested for their response for treatment with fluoroquinolones on HIV-1 [32,33].

We selected a short list of a pro-HIV-1 miRNA (hsa-miR-132-3p [25]), well-established anti-HIV-1 miRNAs (hsa-miR-150, hsa-miR-223, hsa-miR-29a, hsa-miR-28 and hsa-miR-125b [21]), and control miRNAs unrelated to HIV-1 (hsa-miR-142 and -23a), which are strongly expressed in T-cells [48]. To monitor the miRNA expression, we performed miR-qRT-PCR assays on RNA samples in CEM-SS cells (matched to HIV-1 p24 ELISAs, see below) isolated at days 4 and 7 post infectionem (p.i.) treated with Enoxacin and nalidixic acid at each 50 µM standard concentration. Relative miRNA level changes were normalized to DMSO negative controls and U18, Z30 small RNA reference genes. This analysis revealed only a moderate significant reduction of the anti-HIV-1 miR-29a (fc = 0.65; *p* = 0.024) at day 7 p.i (Figure 1a), while all other anti-HIV-1-related miRNAs remained unchanged at both time points. The T-cell miRNA hsa-miR-23a also seemed to be modestly but significantly affected by Enoxacin at day 4 (fc = 0.8; *p* = 0.04). In sharp contrast the levels of the only pro-HIV-1 tested miRNA hsa-miR-132-3p were strongly and significantly reduced to 25% at day 4 p.i. (fc = 0.75; not significant) and 50% at day 7 p.i., (fc = 0.43; *p* = 0.002). A similar effect was seen by a treatment with 100 µM Enoxacin (Appendix A). Together, this points towards a possible connection between miRNA expression level change and anti-HIV-1 effect of Enoxacin. Additionally, we evaluated the effect of nalidixic acid treatment on the same panel of miRNAs as above in CEM-SS cells. In accordance with the negative effect of nalidixic acid both on HIV-1 replication and miRNA expression in CEM-SS, no significant effects or only a very marginal (~10%) hsa-miR-125b induction (fc = 1.12; *p* = 0.03) were observed (Figure 1b and Appendix A). Since there was also no observable change in HIV-1 replication in MT-4 cells by either compound treatment, we omitted to profile the respective miRNA expression herein (Figure 2c and Appendix A). We also measured no effect of nalidixic acid treatment on miR-132-3p expression in CEM-SS cells by qRT-PCR (Appendix A). We therefore focused on the hsa-miR-132-3p pro-HIV-1 Enoxacin interrelation in CEM-SS cells only for the further investigations.

At first, a luciferase reporter assay was designed to answer the question whether we could observe the previously reported RNAi-modulating activity of Enoxacin in CEM-SS cells as well. A psiCHECK reporter plasmid was co-transfected with siRNA targeting renilla (siREN) in the presence or absence of 50, 100 and 150 µM Enoxacin or DMSO control. Indeed, Enoxacin treatment enhanced siREN-mediated suppression of renilla expression (up to three-fold), concluding that a similar RNAi-enhancing effect is present in CEM-SS cells as described (Appendix A, [34]).

### 2.2. Anti-HIV-1 Replicative Effect of Enoxacin but Not Nalidixic Acid in CEM-SS Cells

We next evaluated the effects of Enoxacin and nalidixic acid on HIV-1 replication in CEM-SS and MT-4 cell lines to confirm and correlate our qRT-PCR results of reduced pro-HIV-1 miR-132-3p. Cells were infected with HIV-1 and treated with Enoxacin or nalidixic acid (in 50% DMSO final 50 µM) for up to 10 days. A strong reduction of HIV-1 viral titers (p24 levels) of at least three-fold to up to 78-fold (*n* = 8, *p*-value = 0.0023, two-tailed Mann Whitney U-test) starting from day 7 p.i. was observed—though batch dependent—upon Enoxacin treatment compared to respective DMSO control or untreated cells infected with HIV-1 (=positive control) (Figure 2a,b). In contrast, no significant effects on HIV-1 replication in MT-4 cells were detected upon treatment with Enoxacin or nalidixic acid (Figure 2c). No significant difference was observed between nalidixic acid and DMSO control (Appendix A) nor at 5 µM Enoxacin conditions when CEM-SS and MT4 cells were treated (Appendix A). Moreover, the relative expression changes of hsa-miR-132-3p, but no other exemplary miRNA such as hsa-miR-223 or hsa-miR-23a mirror the anti-HIV-1 effect of Enoxacin (neither nalidixic acid or DMSO treatment) at the time points (day 4 and 7 p.i.) assessed by miR-qRT-PCR (Figure 2d). Since Enoxacin is a proven anti-cancer drug, we intended to rule out whether the treatment eventually flawed its anti-HIV-1 effect in CEM-SS cells by reducing cell viability. Therefore, we performed endpoint cell survival assays after three days treatment using both Enoxacin, nalidixic acid over a concentration range from 10–300 µM and the respective amounts of DMSO negative control (Appendix A). Indeed, no negative growth effect was seen for Enoxacin treatment, while nalidixic acid led to a significant loss of viability (IC50~75–115 µM), with a similar toxic effect seen for high amounts of DMSO (IC50~244–279 µM). Therefore, we conclude that the anti-HIV-1 effect on CEM-SS cells is solely due to Enoxacin administration.

Taken together, we have shown that the fluoroquinolone antibiotic Enoxacin but not nalidixic acid exhibited a strong anti-HIV-1 effect in CEM-SS or in MT-4 cells in vitro, and that it modulated hsa-miR-132-3p expression levels. Due to the absence of any biological effect of nalidixic acid on HIV-1 replication, we continued to study the impact of Enoxacin.

### 2.3. Hsa-miR-132-3p Can Enhance HIV-1 Replication and Eventually Rescue the Anti-Replicative Effect of Enoxacin

Hsa-miR-132-5p has been recently reported to enhance HIV-1 replication and can therefore be defined as a novel favorable host factor for virus reproduction [25]. This phenomenon has been described in Jurkat CD4+ cells and explained by targeting the expression of MECP2, an important context-dependent epigenetic repressor or activator, which is able to repress HIV-1 replication [25,49]. We assessed if hsa-miR-132-3p exhibits a similar enhancing effect on HIV-1 in CEM-SS cells. We did not test this in MT-4 cells because no effect of Enoxacin on miR-132 expression and HIV-1 replication was observed, ruling out any connection. CEM-SS cells were reverse transfected (CEM-SS and MT-4 cells are readily transfectable with small oligonucleotides as monitored by flow-cytometry of fluorophore-labeled oligos; Appendix A) with 100 nM of hsa-miR-132-3p mimic or respective miR-mimic control plus DMSO, or DMSO control and mock transfection, spin-infected with virus and p24 levels measured by ELISA at day 2 p.i. Indeed, we were able to recapitulate a significant HIV-1 enhancing effect of hsa-miR-132-3p as early as day 2 p.i. compared to scrambled control and mock transfection (Figure 3). The p24 levels were increased by two-fold compared to DMSO control mock or scrambled control (*p* = 0.0053 (mock); *p* = 0.004 (scrambled)).

Next, we determined if the anti-HIV-1 effect of Enoxacin was at least partly attributed to reduced pro-HIV-1 miRNA hsa-miR-132-3p levels. To this end, we conducted a rescue experiment by combining Enoxacin treatments (50 µM final concentration) with transfections of hsa-miR-132-3p mimics (100 nM final concentration) or scramble miRNA control and DMSO controls and monitored again the p24 level changes in CEM-SS cells for up to 10 days p.i. No gross cytotoxic stress upon triple treatment could be monitored by trypan blue staining (not shown). Congruent with Figure 3, hsa-miR-132-3p exhibited a slightly HIV-1 enhancing effect at early time points, which was stronger in the presence of Enoxacin compared to DMSO negative controls or scrambled miRNA control transfection. Enoxacin in combination with scramble oligonucleotide transfection showed a potent repressive effect on HIV-1 replication of 5-57-fold (Enoxacin vs. DMSO control) confirming our previous findings. Furthermore, simultaneous application of Enoxacin and hsa-miR-132-3p caused a modest, but not significant reduction in p24 levels compared to the respective miR-scrambled/Enoxacin control combination (Figure 4). This may be in trend indicative that the anti-HIV-1 replicative effect of Enoxacin might at least partially be explained by the down-regulation of pro-HIV-1 hsa-miR-132-3p in CEM-SS cells (Figure 1a,b).

### 2.4. DICER1 Processing of hsa-miR-132 Is Affected by Enoxacin In Vitro

Finally, we aimed to functionally characterize the Enoxacin-dependent expression level changes of pre-hsa-miR-132. We conducted DICER1 cleavage assays [50] of chemically synthesized pre-miR-132 hairpin in vitro in CEM-SS cell lysates. 5′- and 3′ [32P]-labeled RNAs (labeling strategy for 3p strand see Appendix A) were incubated in CEM-SS lysates (hsa-miR-132-5p) or with full recombinant human DICER1 protein alone (hsa-miR-132-3p), in the presence of either 300 and 1000 µM (hsa-miR-132-5p) or 300, 600 and 1000 µM (hsa-miR-132-3p) Enoxacin or an equivalent volume of DMSO control. Cleavage products were resolved by 12% denaturing Urea PAGE and bands corresponding to mature and precursor miRNA quantified to DMSO control (Figure 5a) or as percent cleavage to t = 0 (Figure 5b, see Appendix A for uncropped radiograms, all replicates and individual quantifications). Overall, the results suggest less efficient DICER1 processing of pre-miR-132 in the presence of Enoxacin compared to the DMSO control [51]. Pre-miR-132 processing efficiencies followed a concentration-dependent reduction, with ~50% reduction observed at the highest concentration tested. These results suggest that Enoxacin can inhibit the DICER1 processing of pre-miR-132 in vitro, which could be partially responsible for its anti-HIV-1 effect (Figure 1 and Figure 2).

To gain more mechanistic insights into the observed repression of DICER1 processing we conducted chemical structural probing by SHAPE-MaP [52] of in vitro transcribed pre-miR-132 in the presence of 150 µM, 300 µM Enoxacin and no drug as negative control (Figure 5c and Appendix A). Following ShapeMapper2 analysis, the determined reactivity scores were plotted color-coded onto the pre-miR-132 secondary structure predicted by RNAstructure [53] in each condition. We noted a significantly increased SHAPE reactivity of nucleotides located in the dsRNA stem below the apical loop (nucleotides 19–28 and 43–50, Figure 5c, red box) including the predicted DICER1 cleavage sites for both 150 µM and 300 µM Enoxacin condition (normalized *p*-values = 0.03646 and 0.03997 respectively, Mann-Whitney U test) compared to 0 µM negative control. This could be indicative of a structural re-arrangement of the stem potentially affecting DICER1 recognition and cleavage, being in line with the observed behavior in the DICER1 cleavage assay (Figure 5a,b).

## 3. Discussion

The combination of previous findings illustrating the potential effects of Enoxacin and other fluoroquinolones on enhancing the RNA interference pathway led to our hypothesis that potential anti-HIV-1 replicative or cytoprotective effects could be at least partially attributed to altered levels of miRNAs involved in the modulation of the HIV-1 life cycle [32,33,34,35,45]. Indeed, we identified a strong anti-HIV-1 effect for Enoxacin but not nalidixic acid, which is a structurally related molecule suggesting an Enoxacin-specific mechanism rather than a compound class effect. The differential effects of Enoxacin and nalidixic acid may be explained by their different structures: Both molecules contain the 1,8-naphthyrid-4 on the scaffold substituted in 3-position with a carboxylic acid. Unlike nalidixic acid, Enoxacin contains 6-fluoro- and 7-piperazin-1-yl substituents that are capable of forming additional interactions with target molecules. The anti-HIV-1 effect of Enoxacin turned out to be cell type specific, as it was observed in CEM-SS but not in MT-4 T-cell leukemia cell lines. Interestingly, from all known HIV-1 associated miRNAs tested, the only pro-HIV-1 hsa-miR-132-3p was responsive to Enoxacin treatment only in CEM-SS cell line, while all other miRNAs were insensitive to both drugs in both of the two cell lines. It would be of particular interest to study the T/_2_ differences of miR-132 by Enoxacin versus unaffected miRNAs. However, though a wide variation, most miRNAs display an extraordinary long half-life with a median > 30 h [54,55], which is beyond cell toxicity levels of many commonly used RNA polymerase II inhibitors.

We were able to demonstrate that hsa-miR-132-3p overexpression was in trend able to partially revert the effect of Enoxacin. On a molecular level, Enoxacin specifically altered the pre-miR-132 processing efficiency into 5p and 3p mature strands by DICER1, explaining—at least in part—our results.

HIV-1, as any virus, depends on cellular machineries for its propagation. Consequently, many cellular factors may be supportive or detrimental (e.g., the innate immune system) for virus replication and tend to be relevant in almost all phases of the viral life cycle [56,57,58,59,60]. Whether hsa-miR-132 represents such a bona fide critical host factor, or whether it is simply useful by (re-)shaping the cellular milieu for optimal HIV-1 replication or even if it is negligible, remains an important question for potential future therapeutic applications [25]. Eventually, more miRNA host factors similar in this context could be detected by an unbiased sequencing approach. More specifically, our data raise the question if an analogy may be drawn to hsa-miR-122 and HCV, i.e., if there is a potential to develop a small-molecule drug targeting approach against hsa-miR-132. Furthermore, understanding to what expression level and in which cell type hsa-miR-132 is important for the HIV-1 life cycle requires further testing with a special emphasis on evaluating potential escape responses. Our recently developed whole targetome identification tool miR-CLIP may give additional insight into whether hsa-miR-132 is a critical host factor and therefore a potential attractive therapeutic target [61]. This effort may accomplish to identify additional critical targets, which are involved in the HIV-1 life cycle other than the yet only known gene MeCP2 [25,62].

Our observation that Enoxacin exhibits only a potent anti-HIV-1 effect in one of the tested T-cell lines (CEM-SS) raises the interesting question to what extent this phenomenon is restricted to certain cell types. To address this issue studies, including additional T-cell lines and primary human CD4+ T-cells will be necessary. We have described a partial hsa-miR-132 dependent HIV-1 inhibitory effect of Enoxacin only in CEM-SS, but not MT-4 cells (Figure 2 and Appendix A). MT-4 cells are pre-infected with another human retrovirus HTLV-1 [63], which has been previously shown to suppress RNAi silencing pathway by *Rex* protein interaction with DICER1 [64]. Consequently, the fact that MT-4 cells are insensitive to Enoxacin-mediated modulation of hsa-miR-132 processing may be attributed to a potential counter-effect of HTLV-1 or other cell-intrinsic factors, in which the RNAi machinery cannot be further tuned by Enoxacin, possibly explaining our negative result.

Initially, Enoxacin has been reported to directly bind to the TRBP protein, which is an integral part of the RISC loading complex [43,44,65] increasing its affinity to pre-miRNA or siRNA thus enhancing RNAi by facilitating processing and mature strand loading [34,45]. However, both reports documented a vast majority of unaffected miRNAs upon Enoxacin treatment and out of the differentially expressed ones only 8–14% were induced and some miRNAs (up 3%) were even negatively affected. This indicates that other factors such as RNA structural or sequence-dependent features might play a role in Enoxacin-mediated RNAi modulation or might explain the observed insensitivities to the drug. Enoxacin clearly exhibits no general responses on miRNA processing leaving its specific mechanism(s) undetermined since TRBP seems to be broadly necessary for efficient miRNA processing [44,65]. Nevertheless, these data are in line with our finding that only one out of eight tested miRNAs—namely hsa-miR-132—was influenced by Enoxacin administration. Moreover, our DICER1 cleavage assays showed both in whole CEM-SS lysate containing the complete RISC loading complex and when using only recombinant DICER1 alone a reduced processing of pre-miR-132 to both 5p and 3p strands in vitro. This finding is supported by our SHAPE-Map secondary structure chemical probing, where we determined potential structural rearrangement of pre-miR-132 by Enoxacin to influence DICER1 processing. This is in contrast to the reports that certain miRNAs like pre-let-7 or pre-miR-30a require recombinant TRBP, DICER1 and Enoxacin in a similar in vitro cleavage assay to enhance miRNA processing. Together, this points towards a different and more direct mode of action in the case of pre-miR-132 processing [34]. Indeed, it has been proposed that fluoroquinolones may be capable of directly binding to RNA [66,67,68,69]. For instance, the closely related aminoquinolone WM5 and derivatives bound with up to nanomolar affinity to the HIV-1 Tat RNA hairpin structure [70,71,72]. A direct binding of Enoxacin to pre-miR-132 may inhibit miRNA maturation by DICER1, e.g., by increasing the thermodynamic stability of the hairpin, by preventing pre-miR duplex unwinding or structural alterations to mask recognition by DICER1. Further biophysical studies and/or systematic structure analyses will be needed to answer these questions given the current strong interest in directly targeting RNA structures with small molecules. Enoxacin exhibits gross pleiotropic effects in cells, for instance, its newly discovered involvement in RNA-editing events by binding to adenosine-deaminase 1 (ADAR1) [40]. Thus, we cannot rule out a more complex mode of action other than its effect on pre-miR-132 in relation to our observed anti-HIV-1 activity. Additionally, we only observed a slight partial rescue of hsa-miR-132 on HIV-1 replication. Therefore, it seems clear that the involvement or alterations of other cellular factors or even further miRNAs may account for the outcome of hsa-miR-132-3p overexpression on the anti-HIV-1 effect by Enoxacin (Figure 4), which would require further careful assessments beyond the scope of this study.

## 4. Materials and Methods

### 4.1. Viruses, Cell Lines and Virus Infections

HIV-1HBX2 virus stock was generated by transfection of 293T cells with the HIV-1 full-length plasmid pHXB2, kindly provided by Dr. Marek Fischer, University of Zurich, Switzerland. Virus stock was harvested 48 h post transfection and filtered through a 0.45 μm pore-size filter. CEM-SS and MT-4 cells were infected with HIV-1HBX2 by spinoculation [73] for 2 h at 1200 g at a multiplicity of infection (MOI) of 0.01. Cells were cultivated in RPMI 1640, 10% fetal bovine serum (FBS) 2 mM Glutamine at 5% CO_2_ at 37 °C. The human T-cell leukemia cell lines CEM-SS [74] and MT-4 [63] were obtained through the NIH AIDS Reagent Program, Division of AIDS, NIAID, NIH from Dr. Peter L. Nara and Dr. Douglas Richman, respectively.

### 4.2. Compound Preparation

Enoxacin and nalidixic acid (Sigma-Aldrich, St. Louis, MO, USA) 10 mM stock solutions in 10 mL total volume were prepared freshly the day before usage in cell culture grade 50% DMSO (Sigma-Aldrich, St. Louis, MO, USA) with 5 mM NaOH final concentration under occasional shaking/ultra-sonication. Solutions were kept in the dark overnight at room temperature to allow for complete dissolving. Typically, each 50 μM final standard concentration of compound in 500 μL cell culture media volume had a final concentration of 25 μM of NaOH and 0.25% DMSO, which were well tolerated by cells as tested for viability. Note: All DMSO negative controls in all assays also contained the respective final amount of NaOH.

### 4.3. p24 ELISA with Enoxacin/Nalidixic Acid and Rescue Assay with hsa-miR-132-3p

Assays were performed in 24-well plates with 5 × 105 cells in 500 μL final volume. Up to *n* = 8 for anti-HIV-1 Enoxacin (Figure 2 and Appendix A) and *n* = 2 for hsa-miR-132-2p/Enoxacin rescue assays were performed always in technical duplicates. Cells were infected as described above. 5 and 50 μM final concentrations of Enoxacin or nalidixic acid was added thereafter. Time points for cell culture supernatant subjected to p24 ELISA were collected at days 0, 4, 7, and 10 for Enoxacin/nalidixic acid treatments or at days 0, 3, 5 and 10 for rescue assays, respectively, due to higher cell toxicity upon triple-treatment (virus infection, compound treatment and oligonucleotide transfections). Control treatments were the following: Untreated cells infected with HIV-1, input (=medium), and corresponding volumes of 50% DMSO similar to volumes used for compounds at each concentration. Virus replication was assessed from culture supernatants by p24 ELISA (adapted from [75]. Transfection of pre-miR mimics and controls for rescue assays was performed as described below at a final concentration of 100 nM 24 h before further treatment with compounds and HIV-1 infections. Analysis was performed using GraphPad Prism^®^ 8.0 by *p* value calculation using the Mann–Whitney U test, two-tailed (Enoxacin 50 µM versus DMSO control) while DMSO control = nominator (fold inhibition 1) and fold inhibition = AUC (x) * 100/UC (DMSO control). We did not observe treatment-related cell toxicity throughout our assays monitored by Trypan Blue staining.

### 4.4. Cell Survival Assay

CEM-SS cells were cultivated in biological triplicates for 3 days in a 96-well plate (seeding density 20,000 cells per well) in 0, 10, 50 150 and 300 µM of Enoxacin, nalidixic acid or the respective volumes of DMSO negative control. At day 3, surviving cells were monitored using CellTitreGlo (Promega, Madison, WI, USA) according to the vendor’s instructions. Analysis was performed using GrapPadPrism 8.0 non-linear regression non-linear fit.

### 4.5. MiRNA RT-qPCR

RNA from cell lysates used for Enoxacin HIV-1 ELISA assays was extracted with Trizol ™ (ThermoFisher, Waltham, MA, USA), quantified with Nanodrop (ThermoFisher, Waltham, MA, USA) and Taq-Man miRNA qRT-PCR performed according to the producer’s protocol on a Roche LightCycler 480^®^ device and analyzed by ΔΔCt-method [76] relative to control probe pool (U18 and Z30 small RNA) and DMSO only treated cells.

### 4.6. In Vitro DICER1 Assay in CEM-SS Cell Lysates and with Recombinant DICER

In vitro DICER1 cleavage assays were performed as described by Leuschner and Martinez [50] with modifications. 10 mM Enoxacin stocks were freshly prepared for each experiment as above-mentioned. Hsa-pre-miR-132 was chemically synthesized as described [61]. For radiolabeling, 1 µL of a 1 µM of miRNA working solution was incubated with 1.5 µL ATP ([γ-32P]-6000 Ci/mmol, 10 mCi/mL, Perkin Elmer, Waltham, MA, USA), 1 µL RNase inhibitor (Promega) and 10 units T4 polynucleotide kinase (NEB) in 1× PNK buffer in a total of 10 µL reaction volume at 37 °C for 60 min. For 5′ end labeled pre-miRNAs, the solution was heat inactivated for 5 min at 95 °C. The internal radiolabeling strategy of pre-miRNAs at the 3p strand was described by Schlösser and Hall [51]. The precursor was divided into two fragments, one of which was composed of the 3′ terminus and approximately half of the 3p strand and the second fragment comprised the whole 5p strand, the terminal loop and the remaining half of the 3p strand. The 3′ terminus bearing fragment was likewise isotopically labeled at its 5′-end as before described. Meanwhile, the 5′-terminus containing fragment was non-radioactively labeled at its 5′-end with 32P. For annealing, equimolar concentrations of both fragments were incubated at 95 °C for 5 min and allowed to cool to room temperature for at least 1 h. 10 units of T4 dsRNA Ligase 2 (NEB) were then added to the solution and incubated for 1 h at 37 °C. In order to remove free ATP, the labeled reaction mixtures were column purified using Sephadex G-25 columns (GE Healthcare) according to manufacturer’s recommendations. For in vitro DICER1 assays in CEM-SS cell lysate, the labeled precursors were diluted 1:100 in H_2_O (final concentration 1 fMol/µL) while 1 µL was used for each assay. Cleared cell lysates were harvested in NP-40 lysis buffer (50 mM HEPES, pH 7.5, 150 mM KCl, 0.5% NP-40, 0.5 mM DTT, 2 mM EDTA, complete protease inhibitor EDTA-free (Roche) and 50 U/mL RNase inhibitor) and incubated for 15 min on ice. Concentrations were assessed by A280 and adapted to 12 mg/mL total protein. Aliquoted samples were flash-frozen in N2 and stored at −80 °C. DICER assays were performed in 6 µL cell lysate, 1 µL radiolabeled RNA and the appropriate volume of Enoxacin (for the final concentration of 100 µM: 1:100 stock dilution; 300 µM: 1:33.3 stock dilution; 1000 µM: 1:10 stock dilution) or DMSO. As a control, each RNA was treated in lysate similarly without compounds only. Reactions were performed at 35 °C for 1.5 h and stopped by immediate cool-down on ice. In vitro DICER assays with recombinant DICER protein (OriGene, Rockville, MA, USA) were performed in 5 µL final reaction volume, consisting of 1 µL RNA, the appropriate volume Enoxacin or respective amount of DMSO in DICER reaction buffer (25 mM Tris, pH 7, 25 mM NaCl, 1 mM DTT, 2 mM MgCl2, 1% glycerol). Reactions were performed at 37 °C for 0 min or 1 h and stopped by immediate cool-down. Samples were dissolved in 2x Gel Loading Buffer (Thermo Fisher, Waltham, MA, USA), denatured at 95 °C for 5 min and cooled down on ice prior to centrifugation. Samples were then separated by 12% denaturing urea-PAGE (for assays in cell lysate) or 20% denaturing urea-PAGE (for recombinant DICER assays) in 1× TBE buffer. When the running front reached the bottom of the gel, cover plates were removed and the gel was incubated overnight in a Phosphor-Imager Screen (GE Healthcare, Chicago, IL, USA). The screen was developed on a Typhoon FLA 7000 imager and bands were densitometrically quantified using ImageJ [24]. Normalization of signal intensities for mature miRNAs was performed to DMSO negative control set to one for each concentration relative to pre-miRNA hairpin signal.

### 4.7. SHAPE Assay

SHAPE-MaP (Selective 2′-Hydroxyl acylation Analyzed by Primer Extension and Mutational Profiling) assay consisted of several consecutive steps:(A)Pri-miR-132 In Vitro Transcription

The sequence of pri-miR-132 was taken from miRBase and designed into a dsDNA template. PCR assembly was used to prepare the DNA template with overlapping primers by Primerize web server and followed their protocol [77]. All DNA primer sequences are listed in Appendix A. Fusion High-Fidelity PCR master mix with HF buffer (Thermo Fisher, Waltham, MA, USA) was used to amplify the DNA template, and size checking of the PCR product was performed by agarose gel electrophoresis (1.5% agarose, 1× TBE buffer). The DNA template was purified using a PCR clean-up kit (Qiagen, Minden, Germany). Pri-miR-132 in vitro transcription (IVT) was performed using New England Biolabs (NEB) T7 polymerase and followed the manufacturer’s protocol. Each reaction contained 1 µg/mL of the purified dsDNA template, 0.5 mM of dNTPs, 0.2 M EDTA, 1× NEB reaction buffer (40 mM Tris-Cl pH 7.9, 6 mM MgCl2, 1 mM DTT, 2 mM spermidine), and 100 units of T7 RNA Pol (NEB). The reaction mix was incubated at 37 °C for 1 h. After checking the produced RNA on denaturing Polyacrylamid gel (15% Polyacrylamid, 7 M Urea, 1× TBE—15 min preheating, 30 min running the IVT product at 250 V), DNase I treatment was performed to remove the DNA template, followed by an RNA clean-up step for purification.

(B)Chemical Modification and Mutational Profiling of Pre-miR-132

The published protocol of Smola et al. [52] was followed with a few adaptations. For each experimental condition, at least two technical replicates were used with one negative control with no chemical modifier and one positive control (or denatured control DC) indicating the highest rate of modification for a target RNA. To study the effect of Enoxacin, two different experimental groups with final concentrations of 150 µM and 300 µM of Enoxacin per reaction were applied.

To perform chemical probing, as the first step, pre-miR-132 was re-folded (15 pmol in ddH_2_O) by heating it up at 95 °C for 2 min and snap cooling on ice, again for 2 min. Folding buffer was added to a final concentration of 111 mM HEPES pH 8.0, 111 mM NaCl, 11 mM MgCl2, and the samples were incubated at 37 °C for 30 min. For conditions with Enoxacin, the compound was added to the folding mix right before the 37 °C incubation so it could co-fold with RNA. To reduce sampling error, samples were folded for each experimental condition (which are 2–3 technical replicates, one negative control with no chemical modifier) as a pool and then divided into separate reactions.

After 30 min incubation at 37 °C, 3 µL of 1M7 (1-methyl-7-nitroisatoic anhydride (Sigma-Aldrich, St Louis, USA); i.e., the SHAPE reagent [78]) was added from 100 mM stock, to 9 µL of the folded pre-miR-132 in plus modifier reactions, and the same volume of DMSO (the respective solvent of 1M7) was added to the negative control(s). The reactions were incubated for 2 min at 37 °C. To quench the modification, ddH_2_O was added to reach a total volume of 50 µL per reaction, and then the RNA was recovered by using a G-25 spin column (Cytiva, Marlborough, MA, USA) following the manufacturer’s manual. For the positive controls, 15 pmol of RNA with Enoxacin in DC buffer (50 mM HEPES pH 8.0, 4 mM EDTA) and 50% Formamide *v*/*v* were heated at 95 °C for 1 min, then 3 µL of 1M7 (from 100 mM stock) was added, followed by another 1 min incubation at 95 °C and snap cooling and incubation on ice for one minute.

(C)Reverse Transcription

SuperScript II Reverse Transcriptase (ThermoFisher, Waltham, MA, USA) was used for all reactions. We first annealed 10 µL of the recovered RNA with 6 pmol of reverse transcription primer (Appendix A) in freshly made MaP buffer (50 mM Tris pH 8.0, 75 mM KCl, 6 mM MnCl2, 10 mM DTT, 0.5 mM dNTPs) by heating up samples at 65 °C for 5 min and snap cooling on ice for another 5 min. After adding the enzyme, the samples were incubated at 42 °C for 2 h, followed by 15 min at 70 °C to inactivate the enzyme. The cDNA products were then recovered using a G-25 spin column (Cytiva) following the manufacturer’s manual. The first and second PCR steps to amplify the produced cDNA were exactly followed according to the protocol [52] (for primer sequences, see Appendix A). Generated libraries for next generation sequencing were sent to Novogene Co (Advancing Genomic, Improving Life) with PE150 on an Illumina NovaSeq 5000, yielding at least one million reads per sample.

#### ShapeMapper2 Analysis

The raw data were analyzed using ShapeMapper2, a tool created specifically for the analysis of SHAPE data by Busan and Weeks in 2018 [79]. ShapeMapper2 includes built-in quality control measures and alignment tools that ensure accuracy. Raw sequencing reads were aligned with the target reference sequence -pre-miR-hsa-132-, and experimental conditions were defined-modified, untreated, and denatured samples for comparative analysis. The software produced reactivity data in the form of .shape files, which were further analyzed statistically.

In the analysis, DICER1 regions were specifically taken, corresponding to nucleotide indices 19–28 and 43–50. Sequences GATTGTTACT and TAACAGTC correspond to these sites, respectively. For these segments, the mean reactivity was normalized against the respective mean reactivity of the no treatment conditions so that the real influences of Enoxacin treatment at both 150 µM and 300 µM would be accurately estimated. The Mann–Whitney U test was used to assess significant statistical differences by comparing the reactivity differences between treated versus untreated samples, and Z-values corresponding to the U statistics were calculated to determine the effect sizes. Combined replicates for each drug condition were subsequently input into the RNAstructure tool by Reuter & Mathews, 2010 [53] to visualize the RNA secondary structure based on SHAPE reactivity data. The SHAPE scores were underlined by this analysis, which provided a representation of the level of reactivity at specific positions within the nucleotides.

Command code for ShapeMapper v2.2.0 analysis in R is as follows:

ShapeMapper --name <experiment_name> \

--target <reference_sequence_file> \

--out <output_directory> \

--modified \

--R1 <modified_R1_read_file> \

--R2 <modified_R2_read_file> \

--untreated \

--R1 <untreated_R1_read_file> \

--R2 <untreated_R2_read_file> \

--denatured \

--R1 <denatured_R1_read_file> \

--R2 <denatured_R2_read_file> \

--overwrite \

--output-processed-read \

--output-aligned-reads \

--output-parsed-mutations \

--nproc <number_of_processes> \

--serial

Explanation of Command Components:--name <experiment_name>: Name for the output files.--target <reference_sequence_file>: Path to the reference sequence (FASTA format).--out <output_directory>: Directory for output files.--modified: Indicates modified samples.--R1 <modified_R1_read_file>: Path to R1 read file for modified samples.--R2 <modified_R2_read_file>: Path to R2 read file for modified samples.--untreated: Indicates untreated control samples.--R1 <untreated_R1_read_file>: Path to R1 read file for untreated samples.--R2 <untreated_R2_read_file>: Path to R2 read file for untreated samples.--denatured: Indicates denatured samples.--R1 <denatured_R1_read_file>: Path to R1 read file for denatured samples.--R2 <denatured_R2_read_file>: Path to R2 read file for denatured samples.--overwrite: Allows overwriting existing output files.--output-processed-read: Option to output processed read files.--output-aligned-reads: Option to output aligned read files.--output-parsed-mutations: Option to output parsed mutation files.--nproc <number_of_processes>: Number of CPU threads for processing.--serial: Indicates serial execution (not necessary if using multiple processes).

Analysis Steps in R:Load Data: The script loads and combines data from the specified paths for both drug conditions and the control;Mean Reactivity Calculation: It calculates the mean reactivity for specific indices (19–28 and 43–50) from the loaded datasets;Normalization: The mean reactivity values for both drug conditions are normalized against the mean reactivity of the control condition;Statistical Testing: A Mann–Whitney U test is performed to assess whether there are statistically significant differences between the normalized reactivities of the drug conditions and the control. Z-values are also calculated to evaluate the test results.

Steps for Folding and Annotating RNA with SHAPE Data.

Fold RNA Sequence: ○Open RNAstructure.○Load your .fa (FASTA) file.○Use the RNA Single Fold option.○Generate and save the .ct file.Annotate Structure: ○Go to Annotation Options in RNAstructure.○Input your combined.shape file.Visualize Results:
○Check the annotated structure in RNAstructure.

### 4.8. Flow Cytometry

CEM-SS and MT-4 cells were reverse transfected using RNAiMax™ (ThermoFisher Scientific, Waltham, MA, USA) as described below using Cy3-labeled pre-miR and anti-miR-negative controls #1 (ThermoFisher Scientific AM17120 and AM17011) in 3, 30, and 90 nM final concentrations each or mock transfected. After 24 h, cells were fixed and subjected to flow-cytometric analysis (BD FACS-Aria). Raw data were visualized using FlowJo v10.10 (FlowJo LLC) software.

### 4.9. Luciferase Assay

1 × 104 CEM-SS cells were reverse transfected using RNAiMax™ (ThermoFisher Scientific) according to the manufacturer’s recommendation and seeded in RPMI (ThermoFisher Scientific, Waltham, MA, USA) in 96-well plates. First, transfection of siREN was performed at 30 nM final concentrations after 24 h. Another 24 h later, a second transfection of 20 ng empty dual luciferase reporter plasmid (Promega psiCHECK2) was performed, including a sensor site reverse complementary to the corresponding siREN guide strand, and relative luciferase activity was measured on a Berthold Mithras LB940 Luminometer. The cells were under 50, 100 or 150 µM Enoxacin concentration treatment or respective equivalent volume of DMSO or without treatment. All values were normalized to the level of transfection efficiency measured through the expression of a luciferase not targeted by the siRNA. The error bars represent the s.d. of at least three technical replicates carried out in two independent biological transfections.

### 4.10. Oligonucleotides, miRNA Mimics and TaqMan™ Probes

#### siREN Sense-Strand: GAGCGAAGAGGGCGAGAAAUU

hsa-miR-150-5p TaqMan™ probe (ThermoFisher: # 4427975-000473)

hsa-miR-223-3p TaqMan™ probe (ThermoFisher: # 4427975-002098)

hsa-miR-29a-3p TaqMan™ probe (ThermoFisher: # 4427975-002112)

hsa-miR-125b-5p TaqMan™ probe (ThermoFisher: # 442797-000449)

hsa-miR-142-3p TaqMan™ probe (ThermoFisher: # 442797-000464)

hsa-miR-23a-3p TaqMan™ probe (ThermoFisher: # 442797-000399)

hsa-miR-132-3p TaqMan™ probe (ThermoFisher: # 442797-000457)

TaqMan™ control probes:

Z30 small RNA: (Accession ID AJ007733) (ThermoFisher: # 001092- 4427975)

U18 small RNA (Accession ID AB061820) (ThermoFisher: # 001204-4427975)

Control- and miR-mimics

hsa-miR-132-3p mirVana^®^ miRNA mimic (# 4464066- MC10166)

mirVana™ miRNA Mimic, Negative Control #1 (#4464058)

### 4.11. Statistics

For qRT-PCR and ELISA analysis, a Mann–Whitney U-test was applied (GraphPad Prizm 8.0 calculator).

## 5. Conclusions

In summary, we provide data that in vitro administration of Enoxacin exhibits a repressive effect on HIV-1 replication in the T-cell line CEM-SS, eventually by inhibiting maturation of the pro-HIV-1 hsa-miR-132-3p. This effect of Enoxacin was likely due to affecting the pre-microRNA hairpin structure. The latter finding is highly significant for the rising field of direct targeting of RNA structures with small molecules [80]. Furthermore, Enoxacin may be the progenitor of a new class of anti-HIV-1 drugs eventually suitable to support conventional clinical applications in the future. To this end, unravelling the detailed mechanistic underpinnings of the hsa-miR-132 and Enoxacin interaction will be crucial for future studies.

## Figures and Tables

**Figure 1 ncrna-11-00008-f001:**
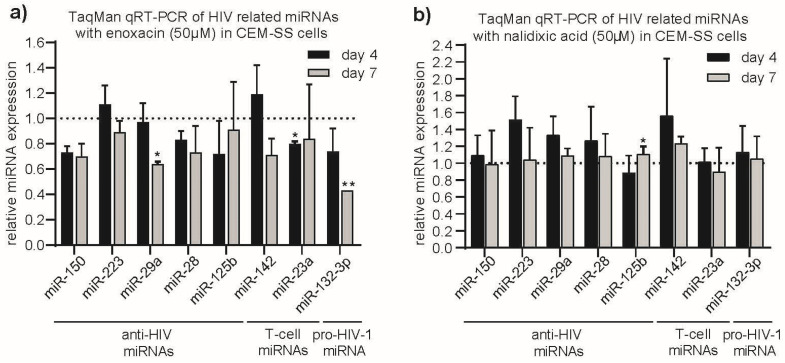
TaqMan qRT-PCR analysis of selected anti- and pro-HIV miRNAs in CEM-SS cells. Cells were treated for 4 and 7 days post-infection with 50 µM final concentration of (**a**) Enoxacin, (**b**) nalidixic acid relative to equivalent volume of DMSO control. * *p* < 0.05 or ** *p* < 0.01 two-tailed Student’s t-test. Error bars indicate ± 1 s.d.

**Figure 2 ncrna-11-00008-f002:**
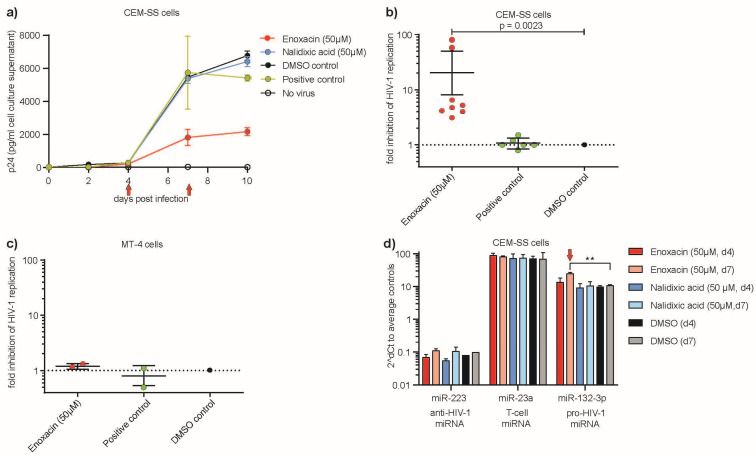
Effect of Enoxacin and nalidixic acid on HIV-1 replication in (**a**) CEM-SS cells over two to ten days p.i. (red arrows indicate time points of qPCR in (**c**)) measured by p24-ELISA. (**b**) Fold inhibition of Enoxacin at 50 µM final concentration on HIV-1 replication compared to controls in CEM-SS and (**c**) MT-4 cells were spin-infected with HIV-1HXB2 at MOI 0.01 and monitored for p24 expression in cell culture supernatant. Untreated cells infected with HIV-1 served as positive, DMSO treatment as negative and no virus as background controls, respectively. (*n* = 4, technical duplicates, DMSO control = nominator, fold-inhibition = 1), ** *p* < 0.01, two-tailed Mann Whitney U test. Error bars indicate ± 1 s.d. (**d**) Pro-HIV-1 miR-132-3p relative expression mirrors the anti-HIV-1 effect of Enoxacin at days 4 and 7 but not control miR-223 (anti-HIV-1) or miR-23-a (T-cell miRNA) compared to DMSO by qRT-PCR. ** *p* < 0.01 two-tailed Student’s *t*-test. Error bars indicate ± 1 s.d., red arrows indicate time points of significance for p24-ELISA and corresponding qPCR.

**Figure 3 ncrna-11-00008-f003:**
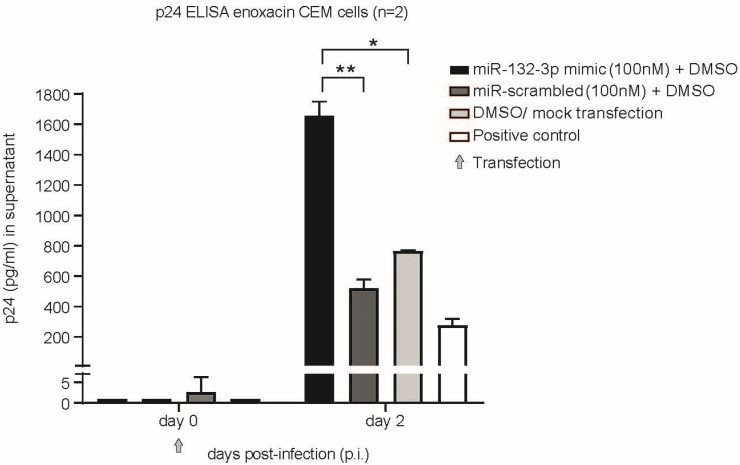
Hsa-miR-132-3p enhances HIV-1 replication in CEM-SS cells. Cells were spin-infected with HIV-1HXB2 at MOI 0.01 and monitored for p24 expression in supernatant at days 0 and 10 post-infection (p.i.) reverse transfected with miR-mimic or scrambled control at 100 nM plus DMSO. DMSO/mock transfection served as negative and cells and virus alone as positive control. Grey arrow indicates time of transfection. *n* = 2, * *p* < 0.05 or ** *p* < 0.01, two-tailed Student’s *t*-test. Error bars indicate ± 1 s.d.

**Figure 4 ncrna-11-00008-f004:**
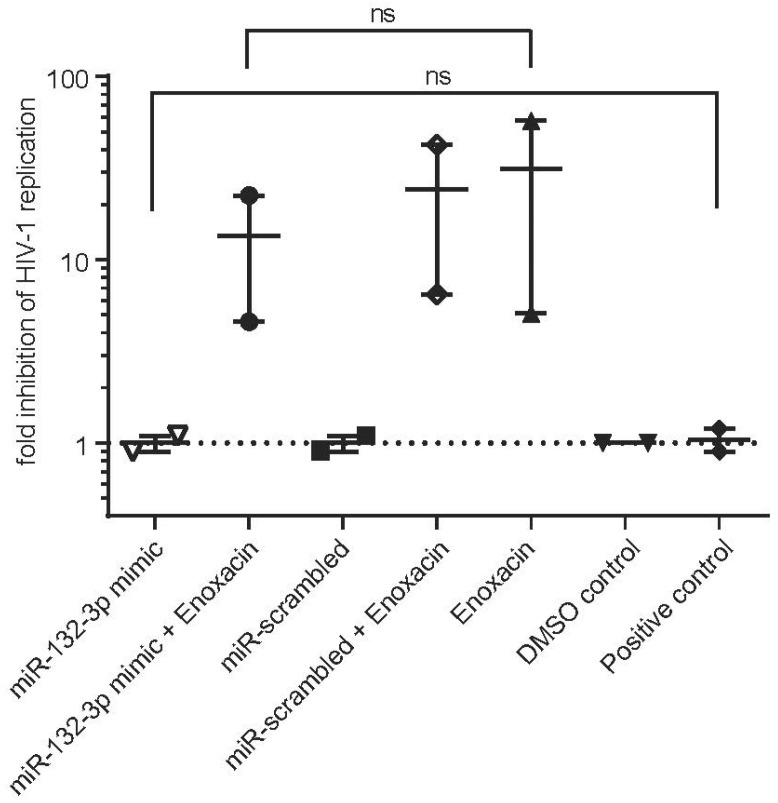
Hsa-miR-132-3p partially rescues Enoxacin-dependent anti-HIV-1 effect in CEM-SS cells. Cells were spin-infected with HIV-1HXB2 at MOI 0.01 and monitored for p24 expression in supernatant over 4 to 10 days post-infection (p.i.) in the presence of 50 µM final concentration of Enoxacin or respective equivalent volume of DMSO control as shown as fold HIV-1 inhibition by p24 ELISA. Untreated cells infected with HIV-1 served as positive control. Cells were reverse transfected in DMSO or Enoxacin treatment with 100 nM concentration of hsa-miR-132-3p mimic or scrambled control. DMSO and Enoxacin treatment with mock transfection served as additional controls. *n* = 2, two-tailed Mann-Whitney U-test, error bars indicate ± 1 s.d.

**Figure 5 ncrna-11-00008-f005:**
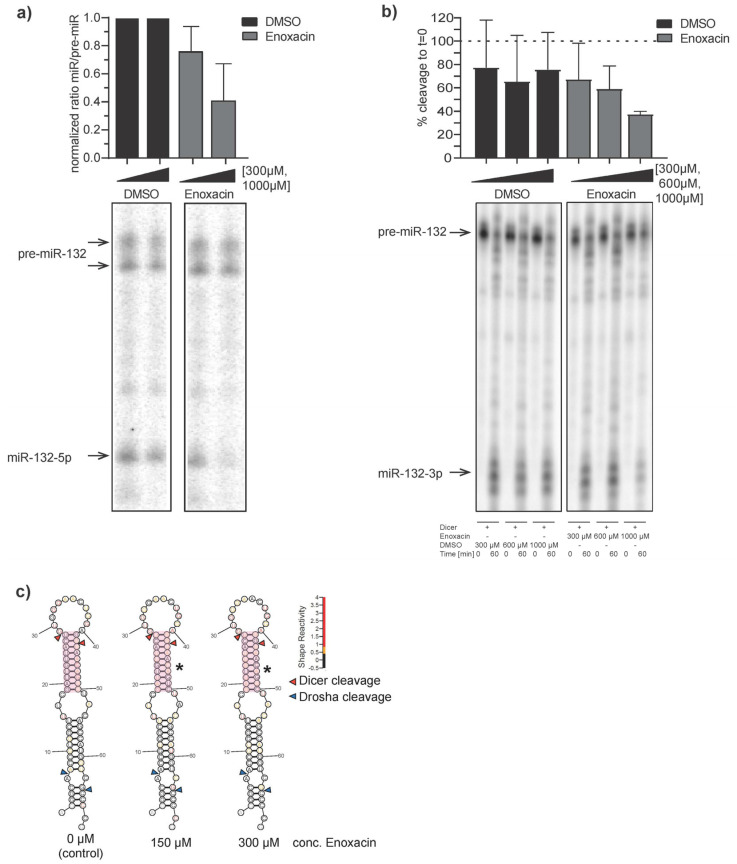
In vitro DICER1 cleavage assay in CEM-SS lysates (**a**) or recombinant DICER1 protein alone (**b**). miR-132 hairpin was 5′- (**a**) or 3′ (**b**) γ-phosphate labeled and incubated in 100 µM, 300 µM and 1 mM Enoxacin or DMSO control at 37 °C for between 0 and 60 min and PAGE-gel separated. Pre-miR and miR bands were densitometrically quantified and DICER1 processing effect was plotted as normalized ratio to DMSO (**a**) or % cleavage (**b**). Note: See full autoradiograms and other replicates Appendix A. (**c**) In vitro SHAPE-MaP reactivity scores plotted on RNAstructure prediction of pre-miR-132 in the presence of 150 and 300 µM Enoxacin and negative control (0 µM Enoxacin). Red triangles indicate DICER1 and blue triangles DROSHA cleavage sites. Light red sequences indicate significant structural rearrangements (marked by asterisks) determined by SHAPE scores of Enoxacin treatment vs. control.

## Data Availability

Source data available on request. SHAPE-MaP sequencing data can be found on Imig, Jochen, 2024, “Anti-HIV-1 Effect of Enoxacin by Modulation of Pro-viral hsa-miR-132 Processing”, https://doi.org/10.17617/3.XLG73X at https://edmond.mpg.de/.

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
