# Peer review of "Anti-HIV-1 Effect of the Fluoroquinolone Enoxacin and Modulation of Pro-Viral hsa-miR-132 Processing in CEM-SS Cells"

_ncrna, 2025, doi:10.3390/ncrna11010008_

Round 1

Reviewer 1 Report

Comments and Suggestions for Authors

This paper has studied the anti-HIV properties of Fluoroquinolone Enoxacin and its potential drug action mechanism. The proposed pro-viral miRNA depression as the antiviral mechanism for Enoxacin is intriguing. The design of the assays was rational and well-thought-out. The conclusion that Enoxacin acts as a cell-type sensitive, pro-has-miR-132 processing modulator was reasonable. The paper has significant meaning in exploring the antiviral mechanisms for enoxacin and its analogs. I suggest its acceptance for the journal. Some minor revisions may be considered for revision before publishing. 1) In line-192, “anti-HIV-1 miR-132-3p” should be “pro-HIV-1 miR-132-3p. 2) Why was the pro-HIV activity of has-miR-132-3p only assessed in CEM-SS cells? 3) In Figure 2, why were the data points in 2c (only 2) far less than in 2b? There are big errors between two data points, with the rest in 2b.

Comments on the Quality of English Language

The English needs polishing. It is borderline readable. 

Author Response

Reviewer #1:

This paper has studied the anti-HIV properties of Fluoroquinolone Enoxacin and its potential drug action mechanism. The proposed pro-viral miRNA depression as the antiviral mechanism for Enoxacin is intriguing. The design of the assays was rational and well-thought-out. The conclusion that Enoxacin acts as a cell-type sensitive, pro-has-miR-132 processing modulator was reasonable. The paper has significant meaning in exploring the antiviral mechanisms for enoxacin and its analogs. I suggest its acceptance for the journal. Some minor revisions may be considered for revision before publishing.

  • In line-192, “anti-HIV-1 miR-132-3p” should be “pro-HIV-1 miR-132-3p.

Authors response: Instead of line 192 we corrected the term anti to “pro-HIV-1 miR-132-3p” in line 182.

  • Why was the pro-HIV activity of has-miR-132-3p only assessed in CEM-SS cells?

Authors response: We aimed to connect the anti-HIV-1 Enoxacin effect (Figure 2a) with a HIV-related miRNA expression pattern change. Since the pro-HIV activity was already assessed in other cell lines (Chiang et al., Virology. 2013), no obvious miR-132-3p expression change upon Enoxacin and anti-HIV-1 replication was observed, we omitted this analysis in MT-4 cells. We added this information to lines 217-218.

3) In Figure 2, why were the data points in 2c (only 2) far less than in 2b? There are big errors between two data points, with the rest in 2b.

Authors response: We observed a wide inter-experimental variation of the anti-HIV-1 effect of Enoxacin in CEM-SS cells possibly due the differences in virus infection efficacies and compound batch preparation. Therefore, we added additional replicates to Figure 2b in order to achieve significance, while it was already obvious after n=2 that in MT-4 there is no affect (Figure 2b/c, see lowest data point in Figure 2b is almost half an order of a magnitude higher than in Figure 2c).

Comments on the Quality of English Language

The English needs polishing. It is borderline readable.

Authors response: We tried to clean-up the English language through-out the text and hope that it is now easier readable.

Reviewer 2 Report

Comments and Suggestions for Authors

Schlösser&Lightfoot et al. present here a manuscript that examines the antiviral effect of Enoxacin, a known fluoroquinolone antibiotic on HSV-1 replication. The authors propose that as Enoxacin can modulate miRNAs pathway, it might have effect on pro-viral miRNAs. Overall this research provides evidence for a novel mechanism by which Enoxacin inhibits HIV-1 replication in CEM-SS cells, however, some flaws and limitations need to be addressed before this reviewer supports the present article for publication. 

Major observations:

  1. The study primarily focuses on CEM-SS cells and finds no effect of Enoxacin on HIV-1 replication in MT-4 cells. This limited scope raises concerns about the generalizability of the findings. The authors acknowledge that MT-4 cells are pre-infected with HTLV-1, which might affect the RNAi machinery, but further investigation into the specific factors contributing to this cell-type specificity is necessary. Investigating Enoxacin's effects in other T-cell lines and primary human CD4+ T-cells is crucial to determine suitability for primary indication.
  2. The study examines a small selection of pro- and anti-HIV-1 miRNAs. A more comprehensive analysis (i.e. NGS of some kind) of the miRNA profile in response to Enoxacin treatment is needed to identify other potential miRNAs that may be involved in its anti-HIV-1 activity.
  3. While overexpression of hsa-miR-132-3p partially rescues the anti-HIV-1 effect of Enoxacin, the rescue is not complete. This suggests that other factors, including additional miRNAs or cellular pathways, may also contribute to Enoxacin's activity. The authors acknowledge this limitation and highlight the need for further investigation into the potential involvement of other cellular factors.
  4. Enoxacin's anti-HIV-1 effect is attributed to its influence on hsa-miR-132-3p processing, which in turn affects HIV-1 replication. However, the exact mechanism by which hsa-miR-132-3p promotes HIV-1 replication remains unclear. Investigating the downstream targets of hsa-miR-132-3p and their roles in the HIV-1 life cycle would strengthen the proposed mechanism. How about the proposed miR-CLIP tool?
  5. Enoxacin exhibits pleiotropic effects in cells, including its interaction with ADAR1, an RNA editing enzyme. These off-target effects may confound the interpretation of the results and raise concerns about potential toxicity. Evaluating the specificity of Enoxacin's interaction with pre-miR-132 and its potential impact on other cellular processes is critical for assessing its safety and therapeutic viability. IC50s vs CC50, Selectivity Index, etc might be some examples of assays authors can consider to address this concerns.

Minor:

  1. It is unusual to see fold inhibition semi-log plots (Figure 2 b and c) where the y-axis was transformed to log. Maybe plot the direct inhibition units would be more formal. 
  2. The same issue happens with the units used as a relative expression (2^dCT) of the miRs. Might need normalized units against gene control. 
  3. For Figure 5c. Shape Reactivity and Shannon entropy are usually plotted for resolved structures.

It seems that this research is preliminary and even when provides compelling evidence for Enoxacin's anti-HIV-1 activity, addressing these flaws will be essential for a more thorough understanding of its mechanism of action and its potential as a therapeutic agent.

Author Response

Reviewer #2:

Schlösser&Lightfoot et al. present here a manuscript that examines the antiviral effect of Enoxacin, a known fluoroquinolone antibiotic on HSV-1 replication. The authors propose that as Enoxacin can modulate miRNAs pathway, it might have effect on pro-viral miRNAs. Overall this research provides evidence for a novel mechanism by which Enoxacin inhibits HIV-1 replication in CEM-SS cells, however, some flaws and limitations need to be addressed before this reviewer supports the present article for publication.

Major observations:

  1. The study primarily focuses on CEM-SS cells and finds no effect of Enoxacin on HIV-1 replication in MT-4 cells. This limited scope raises concerns about the generalizability of the findings. The authors acknowledge that MT-4 cells are pre-infected with HTLV-1, which might affect the RNAi machinery, but further investigation into the specific factors contributing to this cell-type specificity is necessary. Investigating Enoxacin's effects in other T-cell lines and primary human CD4+ T-cells is crucial to determine suitability for primary indication.

Authors response: We are thankful for this critical comment and partially agree. For generalizing an anti-HIV-1 effect of Enoxacin more cell lines and CD4+ T-cells would need to be tested. However, throughout the whole manuscript we refrain from concluding a general anti-HIV-1 effect, but rather always referred to the positively tested cell line CEM-SS. Indeed, we never aimed to proceed to this claim, but instead intended to conduct a proof-of-principle pilot study to connect certain miR-expression changes with Enoxacin treatment and its anti-HIV-1 effect in cell lines. First hints regarding such a potential connection were based on pre-existing literature and we refereed several times to this within the text. We also agree that our current work is far from stating solidly that Enoxacin is suitable “for primary indication” in patients. We consider our work as a potential starting point justifying further mechanistic and in vivo studies whether it may be suited for therapeutics. Therefore, we would refrain to add the requested experimental work and leave it for future studies since we believe that this would be beyond the current scope. However, we consider changing the title as such it suggests limited generalizability e.g.: “Anti-HIV-1 Effect of the Fluoroquinolone Enoxacin and Modulation of Pro-viral hsa-miR-132 Processing in CEM-SS Cells”, but leave this decision open to the editor/reviewer.

  1. The study examines a small selection of pro- and anti-HIV-1 miRNAs. A more comprehensive analysis (i.e. NGS of some kind) of the miRNA profile in response to Enoxacin treatment is needed to identify other potential miRNAs that may be involved in its anti-HIV-1 activity.

Authors response: We clearly appreciate the notion that a more comprehensive in-depth analysis of the miR-expression pattern would be a valuable asset. As we stated (line 146) we aimed to focus on the most established and prominent HIV-related miRNAs. Thus, we prefer to omit such an analysis, also to due current capacity limitations in the labs of both senior authors. However, we added a phrase to discuss (Lines 306-307) that such an approach might be useful to detect other miRNAs of interest in this context.

  1. While overexpression of hsa-miR-132-3p partially rescues the anti-HIV-1 effect of Enoxacin, the rescue is not complete. This suggests that other factors, including additional miRNAs or cellular pathways, may also contribute to Enoxacin's activity. The authors acknowledge this limitation and highlight the need for further investigation into the potential involvement of other cellular factors.

Authors response: We agree and share the idea that other factors apparently seem to be involved, which we concluded from the partial rescue effect of miR-132-3p towards the anti-HIV-1 effect of Enoxacin. However, as there is no real indicative starting point from our data to unravel potential other cellular host factors this opens a wide field of interrogations. We think that these efforts would be beyond the current scope of our work (see also point 4, below).

  1. Enoxacin's anti-HIV-1 effect is attributed to its influence on hsa-miR-132-3p processing, which in turn affects HIV-1 replication. However, the exact mechanism by which hsa-miR-132-3p promotes HIV-1 replication remains unclear. Investigating the downstream targets of hsa-miR-132-3p and their roles in the HIV-1 life cycle would strengthen the proposed mechanism. How about the proposed miR-CLIP tool?

Authors response: The major host factor targeted by miR-132-3p appears to be MeCP2 and was several times mentioned and discussed by us (Chiang et al., Virology, 2013). We share the reviewer’s enthusiasm to unveil further novel host factors being involved in the pro-HIV-1 role of miR-132-3p. We appreciate the suggestion to undertake a miR-CLIP for miR-132-3p, a method introduced by us, eventually undertaken under HIV-1 infection conditions. However, we fairly believe that this effort is far beyond the scope of this manuscript.

  1. Enoxacin exhibits pleiotropic effects in cells, including its interaction with ADAR1, an RNA editing enzyme. These off-target effects may confound the interpretation of the results and raise concerns about potential toxicity. Evaluating the specificity of Enoxacin's interaction with pre-miR-132 and its potential impact on other cellular processes is critical for assessing its safety and therapeutic viability. IC50s vs CC50, Selectivity Index, etc. might be some examples of assays authors can consider to address these concerns.

Authors response: We thank and agree with this point. To substantiate this, we conducted a dose-responsive toxicity assay with Enoxacin and nalidixic acid in CEM-SS cells new Figure S2e). Unfortunately, we do not have the capacity to conduct an analogous assay in MT-4 cell due to currently restricted access to BL-3 safety environment.

Minor:

  1. It is unusual to see fold inhibition semi-log plots (Figure 2 b and c) where the y-axis was transformed to log. Maybe plot the direct inhibition units would be more formal.

Authors response: We prefer to stick to the log scale as this highlights the inter-experimental variation and changes in magnitude fold-change better to the reader. In linear scale data points look very compressed.

  1. The same issue happens with the units used as a relative expression (2^dCT) of the miRs. Might need normalized units against gene control.

Authors response: We appreciate this comment and like to clarify that the 2^dCT are already normalized to the average of the two reference genes U18/Z30, as mentioned in the methods part. By showing the 2^dCT values side-by-side with the DMSO negative control we consider this way of depiction as the most ideal way showing significance at relevant time point for both the miRNA expression and p24-ELISA. We highlighted this in a modified figure with red arrows. Please also see reviewer #3, below.

  1. For Figure 5c. Shape Reactivity and Shannon entropy are usually plotted for resolved structures.

Authors response: We partially agree with this comment. Still, we prefer to keep the current plots onto the predicted structures, as we believe that it will make it easier for the reader to comprehend and compare the region with the strongest re-arrangements and SHAPE-reactivity. We think that the figure, its legend and in the results text were labeled/described sufficiently in order to avoid misunderstandings.

It seems that this research is preliminary and even when provides compelling evidence for Enoxacin's anti-HIV-1 activity, addressing these flaws will be essential for a more thorough understanding of its mechanism of action and its potential as a therapeutic agent.

Authors response: The intricate nature of any kind of research is that it always remains preliminary to some extent opening more questions rather than answering. With this work aimed to address certain focused questions potentially in a very focused manner potentially relevant both to the field of HIV and non-coding RNAs:

  • Are fluoroquinolones and in particular Enoxacin active against HIV-1 in cells via miRNAs?
  • If so, which miRNAs of interest are deregulated by these compounds and why?

We think we fairly addressed these initial questions sufficiently in order to present it to the general public. Still, we are as eager as the reviewer to address more functional mechanistic or a future potential therapeutic application. Nevertheless, we neither aimed for proving a “potential as a therapeutic agent” nor we consider that this would be suited for this publication format.  Instead, we provide first insights of new modality of mode-of-action of Enoxacin on a critical HIV-1 related miRNA, which could be open new points-of-view and future avenues for HIV adjuvant therapeutic approaches.

Additionally, we like to note that the first sentence (“compelling evidence for Enoxacin's anti-HIV-1 activity”) of the last paragraph from reviewer #2 also contradicts itself, as stated in paragraph 1 questioning the generalizability of our date (see top).

We hope that our author´s clarifying responses addressing the raised critical concerns and convince the reviewer and editor that our manuscript is sound for publication in “Non-coding RNA”.

Reviewer 3 Report

Comments and Suggestions for Authors

After entering the host body, HIV will have complex interactions with host genes in order to gain survival advantages. miRNAs are closely related to various physiological functions and pathological processes of the body, as well as to the invasion, latency and progression of HIV, which may provide important references for the development of safer and more effective HIV/AIDS therapeutic drugs.

In the T-cell line CEM-SS, the paper investigated that enoxacin may inhibit hsa-miR-132-3p and thus exert anti-HIV effects, and explored the possible mechanism of action based on the hairpin structure of miRNA.

However, due to the long-term and complex nature of HIV/AIDS, a single alteration in miRNA expression cannot fully reflect the dysfunction of gene function in HIV/AIDS, and more in-depth research is needed to explore the specific function of miRNAs in the disease.

Reading this paper is similar to a maze game, although the author, using perfect logic, eventually gets out of the maze and manages to tell a possible story. In other words, the paper is too many experimental results to interpret, and there are too many different possibilities to understand. For example, the results of hsa-miR-132-3p mimic+Enoxacin in Figure 4 are not significantly different from those of Enoxacin, and it may be precisely impossible to draw the inference of Line239-241.

Restricted to the enormous amount of work that cellular experiments are in reality, and the valuable data from exploratory studies, it is impossible to suggest harsh possibilities for validation in different cells, or perhaps the different results in the two cells in the paper are precisely indicative the fact that drug antiviral effects and miRNAs may not be correlated neither.

However, this could be both a limitation and the value of the paper, in other words, the paper is documenting the results of an experiment on miRNA expression and antiviral effects following drug action in specific cells.

Meanwhile, probably due to the huge workload and long-term data analysis in this paper, although the graphs and charts have been organized beautifully, it is not easy to understand them, for example, Fig. 2b and Fig. 2d, the elevation of scatter plots versus bar graphs actually represents the inhibitory effect of HIV, which may also exacerbate the difficulty of readers' understanding of the explanation of Line 180-182. Similarly, Figure S1c, the inconsistent left-right arrangement of the control histograms and the inhibitory effect after drug action is actually quite a brain-burner with the explanation of the RNAi enhancement effect in Line 187-189.

1. Please add t figure notes for supplementary figures 1-5, which are not visible in the current version, if not for typographical or submission system reasons.

2. Missing figure notes can make it more difficult for readers to understand, for example, the use of the names Nalidixin and Nalidixic acid in Supplementary Figure S2 should preferably be harmonized throughout the paper.

3. Please check if there is a clerical error in Line 192 reduced anti-192 HIV-1 miR-132-3p.

Author Response

Reviewer #3:

After entering the host body, HIV will have complex interactions with host genes in order to gain survival advantages. miRNAs are closely related to various physiological functions and pathological processes of the body, as well as to the invasion, latency and progression of HIV, which may provide important references for the development of safer and more effective HIV/AIDS therapeutic drugs.

In the T-cell line CEM-SS, the paper investigated that enoxacin may inhibit hsa-miR-132-3p and thus exert anti-HIV effects, and explored the possible mechanism of action based on the hairpin structure of miRNA.

However, due to the long-term and complex nature of HIV/AIDS, a single alteration in miRNA expression cannot fully reflect the dysfunction of gene function in HIV/AIDS, and more in-depth research is needed to explore the specific function of miRNAs in the disease.

Authors response: We fully agree with these reviewers´ comment. Of course, HIV/AIDS is a multi-factorial disease and our work in cells cannot be extrapolated to long-term course of disease in patients. Much more scientific work is needed to understand the single factor miR-132-3p in its biological and patient´s context. We partially discussed this aspect (lines 303-306).

Reading this paper is similar to a maze game, although the author, using perfect logic, eventually gets out of the maze and manages to tell a possible story. In other words, the paper is too many experimental results to interpret, and there are too many different possibilities to understand. For example, the results of hsa-miR-132-3p mimic+Enoxacin in Figure 4 are not significantly different from those of Enoxacin, and it may be precisely impossible to draw the inference of Line239-241.

Authors response: We partially agree with this statement that there might be no or only a weak rescue effect of miR-132-3p on Enoxacin anti-HIV-1 activity. That´s why tried to conclude as vaguely as possible (Lines 241-244). However, as we discussed we like to note that the triple treatment of our experimental set-up needs to be tightly balanced and could hinder the determination of an exact cellular outcome (lines 360-364). We leave it up to the reviewer and/or editors to eliminate that phrasing in the final version.

Restricted to the enormous amount of work that cellular experiments are in reality, and the valuable data from exploratory studies, it is impossible to suggest harsh possibilities for validation in different cells, or perhaps the different results in the two cells in the paper are precisely indicative the fact that drug antiviral effects and miRNAs may not be correlated neither.

However, this could be both a limitation and the value of the paper, in other words, the paper is documenting the results of an experiment on miRNA expression and antiviral effects following drug action in specific cells.

Meanwhile, probably due to the huge workload and long-term data analysis in this paper, although the graphs and charts have been organized beautifully, it is not easy to understand them, for example, Fig. 2b and Fig. 2d, the elevation of scatter plots versus bar graphs actually represents the inhibitory effect of HIV, which may also exacerbate the difficulty of readers' understanding of the explanation of Line 180-182. Similarly, Figure S1c, the inconsistent left-right arrangement of the control histograms and the inhibitory effect after drug action is actually quite a brain-burner with the explanation of the RNAi enhancement effect in Line 187-189.

Authors response: We thank the reviewer for this comment and apologize for these inconveniences. We highlighted the current arrows into “red” to point towards significantly changed time points in both the p24-ELISA and respective qRT-PCR and added a note to the legends. In Figure S1c. Indeed, the days 4 and 7 were mislabeled and have been corrected. We hope that these changes will make the readers comprehension easier.

  1. Please add t figure notes for supplementary figures 1-5, which are not visible in the current version, if not for typographical or submission system reasons.

Authors response: We thank the reviewer for this note and added the figure notes as suggested.

  1. Missing figure notes can make it more difficult for readers to understand, for example, the use of the names Nalidixin and Nalidixic acid in Supplementary Figure S2 should preferably be harmonized throughout the paper.

Authors response: We harmonized the term nalidixic acid in Figure S2.

  1. Please check if there is a clerical error in Line 192 “reduced anti-192 HIV-1 miR-132-3p”.

Authors response: We corrected this according to reviewer #1, point 1.

Round 2

Reviewer 2 Report

Comments and Suggestions for Authors

Agree with the authors that the title (and scope) should be modify as authors propose for "Anti-HIV-1 Effect of the Fluoroquinolone Enoxacin and Modulation of Pro-viral hsa-miR-132 Processing in CEM-SS Cells" as it is more specific on their observations and generates a narrower conclusion based on their findings. 

Also, I would like to clarify, in their response what they believe is a contradiction, it seems that authors are not understanding their study limitations. In my opinion they provided evidence that Enoxacin inhibits HIV-1 replication in CEM-SS cells (as I stated in my first review: compelling evidence for Enoxacin's anti-HIV-1 activity), however as I mentioned in the same review 1)they have contradictory results when using MT-4 cells as a model. 2) the research is bias, as small selection of miRNAs was targeted. 3) they failed to provide a mechanism when attempting to rescue in the hsa-miR-132-3p towards the anti-HIV-1 effect of Enoxacin, as a partial rescue effect could be underlying other mechanisms. 3) they need to evaluate off-target effects that might bring unwanted toxicity. I suggested to evaluate the specificity of Enoxacin's interaction with pre-miR-132 and its potential impact on other cellular processes , as it is critical for assessing its safety and therapeutic viability. IC50s vs CC50, Selectivity Index, etc. might be some examples of assays authors can consider addressing these concerns. Any CRO that have a BSL3 would be able to help them with this research.

Overall, this second version provides evidence for Enoxacin as an inhibitor of HIV-1 replication in CEM-SS cells, however, authors need to specify the fllaws and limitations ofthe research by adding "CEM-SS Cells".

Author Response

We thank the reviewer´s comment and clarifications. Now we better understand the statements and fully appreciate our study limitations and adapted the title accordingly. We would like to briefly respond to the points above:

  1. Indeed, we appreciate the cell type selectivity of the Enoxacin´s effect and also discussed this.
  2. It is correct that we only used a small panel of miRNAs and an extension by sRNA sequencing profiling would be more beneficial. However, many other non-RNA based factors may still play a role.
  3. Indeed, we leave over deeper mechanistic of the hsa-miR-132 rescue studies to future research.
  4. We added a new supplemental figure addressing this concern in Figure S2e for CEM-SS cells to best of our capacities. However, both funding and infrastructure access limitations permit an analysis und BLS3 level including CRO.